# Development of Oleogel-Based Fat Replacer and Its Application in Pan Bread Making

**DOI:** 10.3390/foods13111678

**Published:** 2024-05-27

**Authors:** Sung-Huo Kim, Yeon-Ji Jo, Sung Ho Lee, Sung-Hoon Park

**Affiliations:** 1Department of Food & Nutrition, Gangneung-Wonju National University, Gangneung 25457, Republic of Korea; rlatjdgn0616@gwnu.ac.kr; 2Haeram Institute of Bakery Science, Gangneung-Wonju National University, Gangneung 25457, Republic of Korea; 3Department of Marin Bio Food Science, Gangneung-Wonju National University, Gangneung 25457, Republic of Korea; joyeonji@gwnu.ac.kr; 4SPC Group Research Institute of Food and Biotechnology, Seoul National University, Seoul 08826, Republic of Korea; sungholee@spc.co.kr

**Keywords:** oleogels, shortening, white pan bread, bakery products, hydrocolloids

## Abstract

In recent years, the bakery industry has been exploring alternative fats to replace traditional solid fats. Shortening, a common baking ingredient, is produced through the hydrogenation of vegetable oils, resulting in high levels of saturated and trans fatty acids, despite its vegetable oil origin. The excessive consumption of these fats has been associated with negative health effects, including dyslipidemia and cardiovascular issues. Oleogels, incorporating hydroxypropyl methylcellulose (HPMC), xanthan gum (XG), and olive oil, were utilized to replace shortening in the production of white pan bread. The substitution of shortening with oleogel in the white pan bread preparation demonstrated potential reductions in saturated fat, trans fat, and the ratio of saturated fat to unsaturated fatty acids. Specifically, with the complete substitution of shortening with oleogel, saturated fatty acids decreased by 52.46% and trans fatty acids by 75.72%, with unsaturated fatty acids increasing by 57.18%. Our findings revealed no significant difference in volume between bread made with shortening and bread with up to 50% shortening substitution. Moreover, when compared to bread made with shortening and 50% oleogel substitution, no adverse effects on the quality characteristics of volume and expansion properties were observed, and the retrogradation rate was delayed. This study suggests that incorporating oleogels, formed with hydrocolloids such as HPMC and XG, to replace shortening in bread, in conjunction with traditional solid fats, provides positive effects on the quality and nutritional aspects of the bread compared to using oleogel alone. Through this study, we demonstrate the use of oleogels as a healthier alternative to shortening, without reducing the bread’s quality, thus offering a practical solution to reduce unhealthy fats in bakery products.

## 1. Introduction

Numerous ingredients and additives play crucial roles in the baking process. Among them, fats such as butter, shortening, or margarine are particularly important elements that significantly influence the texture, structure, freshness, and flavor of the finished product. Shortening, a semisolid, plastic fat, is composed entirely of fats derived from refined vegetable oils. It can be produced by hydrogenation and also interesterification. Its primary functions include aiding air entrapment, providing a smooth texture, and contributing to a rich flavor [1].

In yeast-leavened products, a tough texture is commonly observed due to the cohesion of starch particles and gluten proteins. Shortening plays a pivotal role in this context by disrupting the protein and starch structures, lubricating the gluten, and ultimately resulting in soft bakery products [2].

However, the shortening and margarine commonly used in the baking process contain high levels of saturated fats and trans fats, which can have negative effects on human health. The excessive consumption of trans fatty acids, in particular, is reported to increase low-density lipoprotein cholesterol levels in the body, negatively affecting cholesterol metabolism and posing a risk of diabetes and cardiovascular diseases [3,4,5]. Moreover, the use of vegetable oils in baking, while rich in unsaturated fatty acids, can lead to reduced product processability and storage stability [6]. 

In recent years, there has been a concerted effort to explore alternatives that allow for a reduction in fat content while preserving product texture and quality without significant changes in chemical composition [7]. One notable avenue in these efforts involves research on using oleogels as a substitute for shortening. Oleogels, characterized as thermally reversible, three-dimensional structures exhibiting viscoelastic behavior, trap organic liquids using low-molecular-weight and high-molecular-weight oleogelators. In comparison to liquid oils, they provide enhanced oxidation stability, improved texture, spreadability, increased processing suitability, and extended storage stability, making them a promising choice for fat replacement in the food industry [8,9,10]. 

Hydrocolloids, water-soluble polysaccharides with various functional properties, are widely used in food technology. A recent innovative method has been proposed to produce oleogels using hydroxypropyl methylcellulose (HPMC) for applications in baking products [5]. Hydrocolloids induce structural changes in the major components of flour during both the processing and storage stages of the baking process, notably enhancing storage stability and quality [11,12,13,14]. 

HPMC, a versatile polymer, reduces interfacial tension in both hydrophilic and hydrophobic liquid interfaces [15,16]. Notably, cellulose derivatives including HPMC, obtained by adding methyl and hydroxypropyl groups to cellulose, exhibit uniform properties compared to the variable nature of hydrocolloids from natural sources. Additionally, HPMC retains partial hydrophilicity of cellulose despite the presence of hydrophobic groups [12,17].

In baking, cellulose derivatives like HPMC are reported to enhance moisture absorption and impede protein binding and the aggregation of starch and gluten, thereby improving product quality by reducing retrogradation, increasing product volume, and providing a soft crumb texture [18,19,20,21,22].

Xanthan gum (XG) is another hydrocolloid commonly used as a thickening agent, stabilizer, and, in baking, as a dough enhancer and quality improver in gluten-free bakery products, within the food industry [23,24,25]. The inclusion of gums is known to augment the viscosity of starch and affect starch gelatinization and retrogradation [26]. Moreover, previous research has confirmed that using XG and gelatin in the formation of oleogels contributes to the stabilization of emulsions through protein–polysaccharide interactions at the oil–water interface [27]. In other studies, the combination of XG and HPMC in oleogel formation was observed to result in the formation of stable emulsions during oleogel formation and demonstrated a degree of stability allowing for moisture removal during the drying process [28].

In this study, oleogels composed of olive oil, HPMC, and XG were formulated to assess their potential and applicability as shortening substitutes in bread production. It is anticipated that the findings will contribute to supporting better food choices for consumers and fostering advancements in the realm of alternative ingredients in the baking industry.

## 2. Materials and Methods

### 2.1. Materials

For the preparation of oleogels, the following ingredients were used: olive oil (extra virgin, Beksul CJ Co. Ltd., Incheon, Republic of Korea), HPMC (AN6, ES Food, Seongnam-si, Republic of Korea), and XG (80MESH, ES Food, Seongnam-si, Republic of Korea). In the preparation of white pan bread, the ingredients included 100 g of wheat flour (Mildawon Co. Inc., Gongju, Republic of Korea), 6 g of refined sugar (CJ CO. Inc., Seoul, Republic of Korea), 6 g of shortening (Samyangsa Co. Inc., Incheon, Republic of Korea), 1.5 g of salt, and 6 g of yeast (Saf-instant gold, Lesaffre Pte Ltd., Marcq-en-Baroeul, France).

### 2.2. Preparation of Oleogel

The oleogels were prepared with modifications to the methods described previously [29,30]. Initially, 2 g of HPMC was dispersed in 76.8 g of cold water and mixed using a food processor (Vitamix E310, Vita-Mix, Cleveland, OH, USA). Subsequently, 1.2 g of XG was added to the resulting aqueous solution and mixed for 5 min. Following this, 120 g of oil was gradually added and homogenized in the processor for 5 min. The resulting emulsions were spread on a Teflon tray using a silicone pastry bag and a plastic nozzle. Drying of the emulsions was achieved using forced convection air in a dry oven (WOF-W155, DAIHAN Scientific Co., Ltd., Wonju, Republic of Korea) at 80 °C for 4 h. This duration was determined as the minimum time required to reach a constant dry weight, maintaining a moisture content of 1.75% ± 0.51%, under the specified conditions. Finally, the dried products were ground in a grinder for 4 s to produce the oleogels.

### 2.3. Preparation of White Pan Bread with Oleogel

White pan bread was prepared using a slightly modified AACC (American Association of Cereal Chemists) standard straight dough method [31]. The ingredients were initially mixed at low speed for 3 min and then at high speed for 7 min using a SK-20 Mixer (SKMixer Co., LTD. Saitama, Japan). The mixed dough underwent a fermentation period of 1 h at room temperature (25 °C) with a humidity of 75–80%, followed by dividing the dough into 380 g portions, setting, shaping, and proofing at 38 °C with humidity ranging between 85% and 95% for 50 min. Subsequently, the bread was baked at 175 °C for 35 min. After baking, the finished product was cooled to an internal temperature of either 25 °C or 35 °C and then packaged in a polyethylene bag. The experimental groups were created by substituting 0% (S100), 50% (S50/O50), and 100% (O100) oleogel instead of shortening. All experiments were conducted in triplicate.

### 2.4. Proximate Analysis of Oleogel Bread

The moisture content of oleogel bread was analyzed using an infrared moisture determination balance (FD-720, KETT Electric Laboratory, Tokyo, Japan). For each sample, 3 g was placed on an 8 cm diameter aluminum foil dish, and measurements were taken at 105 °C for 8 min. The determination of ash, crude protein, and crude fat content in the oleogel bread samples followed the methods recommended by AOAC International [32]. General component analysis of the oleogel bread involved determining the carbohydrate content, calculated by subtracting the combined amounts of moisture, crude protein, crude fat, and ash from 100 g of the sample. The results of the general component analysis in food are typically expressed as percentages [33].

### 2.5. Baking Loss Measurement

The baking loss was calculated using the following equation, which represents the ratio of the weight difference between the dough before and after baking:(1)$$Baking\;Loss\;(\%)=\frac{Before\;Baking\left(g\right)-After\;Baking\;(g)}{Before\;Baking\;(g)}\times 100$$

### 2.6. Specific Volume of Oleogel Bread

The specific volume of white pan bread was measured using a laser scanning system (Volscan Profiler-VSP600, Stable Micro System Ltd., Surrey, UK) following the AACC-approved method (10.00–16.01). Initially, samples of white pan bread were prepared for measurement.

### 2.7. Crumb Structure Analysis of Oleogel Bread

The crumb structure analysis of the bread followed the method outlined in [34]. Sliced bread, with a thickness of 18 mm, underwent scanning using an image scanner (Samsung SL-C1454FW, Republic of Korea), and subsequent analysis was performed using ImageJ analysis software version 1.49 (NIH, Bethesda, MD, USA). The color image was converted into an 8-bit grayscale binary image, and the image resolution was set to 300 dpi. A field of view measuring 30 × 30 mm from the center of the slice was selected and segmented using a threshold to determine metrics such as cell density (cells/cm^2^), average cell area (ratio of pore area to the number of cells), and porosity (ratio of pore area to 3 cm^2^).

### 2.8. Color Analysis of Oleogel Bread

The crust and crumb colors of the white pan bread were assessed using a chromameter (CR-400 KONICA MINOLTA, Tokyo, JAPAN). Color values were quantified in terms of L* (lightness/darkness), a* (redness/greenness), and b* (yellowness/blueness). The browning index was employed to illustrate the color alteration during the baking process [35].

### 2.9. Texture Profile Analysis of Oleogel Bread

Texture profile analysis (TPA) was performed on 18 mm thick slices of bread using a CTX Texture Analyzer (AMETEK Brookfield, MA, USA). The TPA utilized a cylindrical probe with a diameter of 50 mm, a crosshead speed of 60 mm/min, and a strain of 50%. Measurements were taken on day 1, 2, and 4. The texture parameters determined were hardness (N), springiness (mm), cohesiveness, and chewiness (J).

### 2.10. Fatty Acid Composition Analysis of Oleogel Bread

The fatty acid composition of bread prepared using oleogel was determined following the AOAC 996.06 method [36], with triundecanoin (C11:0) serving as an internal standard. The procedure involved acid hydrolysis using 8.3 M HCl for fat extraction. The homogenized samples underwent acid hydrolysis, followed by fat extraction using organic solvents (diethyl ether and petroleum ether). Subsequently, methylation was carried out by adding 0.5 N NaOH in MeOH and 14% BF3 in MeOH. Fatty acid methyl ester analysis was conducted using GC-FID (Hewlett-Packard 6890, Agilent Technologies, Palo Alto, CA, USA) with an SPTM-2560 capillary column (100 m × 0.25 mm ID × 0.2 μm thickness, Supelco, Bellefonte, PA, USA). Helium served as the carrier gas at a flow rate of 0.75 mL/min, with a split ratio of 200:1. The injector temperature was maintained at 225 °C, and the detector temperature at 285 °C. The oven temperature programming included an initial temperature of 100 °C held for 4 min, followed by a gradual increase at a rate of 3 °C/min up to 240 °C, where it was maintained for 15 min. The obtained data were compared to bread prepared using shortening.

### 2.11. Statistical Analysis

Statistical analysis was performed using the SPSS statistics program, version 25 (IBM, New York, NY, USA). To identify significant differences (*p* < 0.05), a Tukey’s multiple-comparisons test within a one-way analysis of variance was employed. The reported values are presented as mean ± standard deviation, with distinct letters indicating significant differences (*p* < 0.05). The data reported in all of the tables are average values of triplicates. 

## 3. Results and Discussion

### 3.1. Physicochemical Quality Characteristics of Oleogel Bread

The external structure and physicochemical quality characteristics of the fat-substituted bread are presented in Table 1 and Table 2, and Figure 1. Volume measurement is a crucial factor in evaluating bread quality [37]. Specific volume analysis revealed no significant difference between the S100 (5.32) and S50/O50 (5.35), whereas O100 (4.85) exhibited a lower value. Additionally, the baking loss rate analysis showed no significant difference between S100 (15.83) and S50/O50 (15.74), but O100 (13.42) exhibited a notably higher difference with increased replacement. The moisture content showed no significant difference between S100 (33.50) and S50/O50 (33.63), but O100 (35.26) displayed a significantly higher moisture content.

These results suggest that substituting with 50% oleogel does not have a major impact on the volume properties and moisture content of the bread. However, O100 results in increased moisture content and decreased volume characteristics. These results may be attributed to the higher content of XG added with increasing levels of oleogel, which, as indicated in [12,38], holds excess moisture and prevents proper oven spring, leading to decreased volume characteristics.

### 3.2. Crumb Structure of Oleogel Bread

The characteristics of the internal structure, such as the crumb structure of bread with oleogel substitution, are shown in Table 1 and Figure 2. The size and number of cells inside the crumb significantly influence the texture and sensory results of the bread [34,39]. The stability of the gluten–starch matrix plays a crucial role in preventing unevenness and adhesion within expanding gas cells. Ensuring the stability of gas cells is contingent upon the ability of the dough to prevent collapse [40]. The analysis of the crumb structure of the bread reveals a consistent pattern with increasing levels of oleogel replacement. Specifically, higher oleogel replacement levels correspond to an increase in the average cell area (total area/cells) and porosity (total area/cm^2^), along with a decrease in cell counts and cell density. The porosity percentages were 0.15% for S100, 0.19% for S50/O50, and 0.30% for O100. This indicates that greater oleogel substitution led to significant alterations in crumb structure, influencing the cell size, density, and porosity. 

The observed decrease in volume may be attributed to the substantial amount of XG, which enhances dough elasticity and viscosity, as reported in previous studies [38,41]. The increased dough elasticity and firmness hinder gas cell expansion, resulting in thicker and harder air cell walls. This affects the stability and structure of the gluten–starch matrix in the crumb, leading to fewer mall cells and a more even crumb structure, consistent with findings in the literature [42,43]. Consequently, the addition of XG results in a more open crumb structure, as demonstrated by a previous study [44]. These findings suggest that similar effects on crumb structure are observed even with 100% oleogel replacement in bread production.

### 3.3. Color Analysis of Oleogel Bread

The color analysis of oleogel bread examined the impact of replacing shortening with oleogel on the bread’s color (Table 3). Comparing the S100 group with the S50/O50 and O100 groups, a decrease in the L* value of the crust was observed in the S100. However, visual examination (Figure 1) did not reveal distinct differences in crust color.

In the crumb, except for the b* value, the L* and a* values showed similar results between the S100 group and the S50/O50 group. However, an increase in oleogel substitution was associated with a significant increase in the b* value, indicating a stronger yellow color. The O100 group exhibited a significant decrease in L* and a* values, while the b* value showed the highest increase. This trend was also evident in the bread’s appearance (Figure 2), where an increase in oleogel replacement was linked to a more pronounced yellow color in the crumb.

The effect of oleogel on bread color was shown by a decrease in the L* value in the crust upon 100% replacement and a significant increase in the b* value in the crumb, indicating a more pronounced yellow color.

These results may be attributed to the color of the added olive oil, which becomes more pronounced with an increase in the replacement amount of oleogel. Moreover, it is worth noting that the fatty acid composition of oleogel bread, differs from that of traditional shortening-based bread. While shortening typically has a white color and is primarily composed of 16:0 saturated fatty acids, oleogels produced from olive oil have a higher content of 18:1 unsaturated fatty acids. This distinction in fatty acid composition may contribute to the observed differences in color parameters.

### 3.4. Texture Profile Analysis of Oleogel Bread

The bread, with shortening replaced by oleogel, was preserved for 4 days, during which texture measurements were conducted. The results are shown in Table 4. A consistent increase in hardness during storage was observed across all groups, irrespective of oleogel substitution. Specifically, the O100 group exhibited the highest hardness measurement (8.43) on day 1, and this trend continued on the fourth day, with O100 recording the highest measurement (14.00). These results stem from the inclusion of XG, as explained by [38] previously, during the production of bread by completely replacing oleogel. This addition of XG leads to increased moisture content, reduced oven spring, and the weakening of the gluten structure. Additionally, solid fats play a vital role in stabilizing bubbles at the air–matrix interface [45]. When shortening is entirely replaced with oleogel, the challenges arise in replicating the mechanism of stabilizing bubbles, a characteristic of solid fats. This could accelerate the reduction in expansion and weakening of the gluten structure, resulting in higher hardness and lower volume.

On the contrary, in the S50/O50 group, retrogradation was delayed the most. In the S100, hardness increased by 99.61% (5.16) on day 1 (5.18) and day 4 (10.34). In contrast, the S50/O50 group exhibited an increase of 77.95% (3.33) on the day 1 (4.40) and day 4 (7.83). These findings indicate that the S50/O50 showed a 35.46% (1.83) delayed rate of hardness increase compared to the S100. Moreover, the substitution of 50% of shortening with oleogel not only led to decreased bread hardness, but also suggested delayed retrogradation. Notably, the S50/O50 group showed lower values of hardness and chewiness, although their specific volume and moisture content values did not differ significantly from those of the S100. These results indicate a positive effect of oleogel on the product without compromising quality when replacing 25–50% of solid fat [5,46]. Additionally, these results indicate that the characteristics of HPMC hydrocolloid, known for reducing bread hardness, manifest when applied up to 50% in the production of bread with oleogel. Substituting oleogel with other baking fats, rather than using oleogel alone, appears to have positive effects on hardness and retrogradation. Although springiness and cohesiveness did not show significant differences with the addition of oleogel, chewiness exhibited notable differences (S100: 34.54, S50/O50: 29.17). Considering that hardness and chewiness are negatively correlated with bread quality, these observations are consistent with previous findings on the effects of bread quality [47,48,49]. Future studies should explore the mechanisms by which oleogels influence these textural properties to further understand their potential benefits in baking applications.

### 3.5. Fatty Acid Composition of Oleogel Bread

The comparison of fatty acid compositions in breads produced by replacing shortening with oleogel is shown in Table 5. The primary fatty acid composition of olive oil consists of 21.9% saturated fatty acids and 78.1% unsaturated fatty acids, with oleic acid (18:1, 68.4%) being the predominant fatty acid [50]. In the O100 group bread, oleic acid (18:1, 61.24) is the predominant component, with saturated fatty acids at 23.80%, unsaturated fatty acids at 75.76%, and trans fatty acids at 0.42%. While these may be slight variations based on olive oil types, the fatty acid composition of this study’s olive oil, used in oleogel and bread preparation, did not show significant changes.

For the S100 group bread, saturated fatty acids comprise 50.06%, unsaturated fatty acids account for 48.20%, and trans fatty acids make up 1.73%. In contrast, the O100 group showed a 52.46% decrease in saturated fatty acids (to 23.80%), a 75.72% decrease in trans fatty acids (to 0.42%), and a 57.18% increase in unsaturated fatty acids (to 75.76%) compared to the S100 group. Consequently, when applying oleogel prepared using HPMC, XG, and olive oil as a replacement for shortening in bread production, it is shown that a nutritionally superior bread with a high level of unsaturated fatty acids, along with reduced levels of saturated and trans fatty acids, can be produced without significant alterations to the overall fatty acid composition. These results provide insights into how the type of oil and fatty acid composition used in oleogel preparation influence the fatty acid composition of bread. Furthermore, this understanding can enhance the potential benefits of incorporating oleogels into bakery products.

## 4. Conclusions

In this study, oleogel was produced by utilizing the hydrocolloid properties of HPMC and XG, combined with unsaturated fatty acid-rich, plant-based olive oil, to replace shortening in bread formulation. The incorporation of HPMC and XG during oleogel production exhibited intrinsic properties that enhanced crumb softening and dough stability, preserving the hydrocolloid attributes even when used in the production of oleogel. In particular, a 50% replacement revealed reduced hardness and retrogradation compared to the control group. Additionally, substituting shortening with 100% oleogel produced using the developed methodology resulted in a significant reduction in saturated and trans fatty acids, enabling the production of bread rich in unsaturated fatty acids. Substituting up to 50% of shortening with oleogel resulted in high-quality bread with reduced levels of saturated and trans fatty acids, without a significant loss of volume or texture compared to the control group. The adjustment of oleogel content facilitated the control of saturated and trans fatty acid content while enhancing bread quality. Challenges associated with fat replacers and oleogels encompass volume reduction, texture alterations, and sensory impacts. However, through the incorporation of HPMC and XG to create oleogel, the appropriate application in bread formulation led to the production of improved bread quality with controlled levels of saturated and trans fatty acids, thereby enhancing both preference and nutritional aspects. 

In conclusion, this study demonstrates that utilizing oleogel produced with HPMC and XG offers the potential to create high-quality bread with a controlled fatty acid composition and improved sensory and nutritional attributes, providing an effective approach to tackle the challenges of fat replacers and oleogels in bread formulation. However, a higher content of unsaturated fatty acids may impact oxidative stability and shelf life, necessitating further research on their effects on preservation and storage conditions.

## Figures and Tables

**Figure 1 foods-13-01678-f001:**
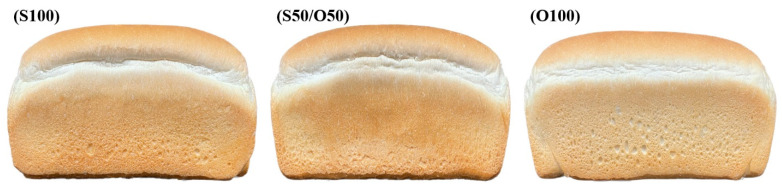
The general shape and appearance of pan bread containing oleogels: (S100) control; (S50/O50) oleogel with 50% shortening substitution; (O100) oleogel with 100% shortening substitution.

**Figure 2 foods-13-01678-f002:**
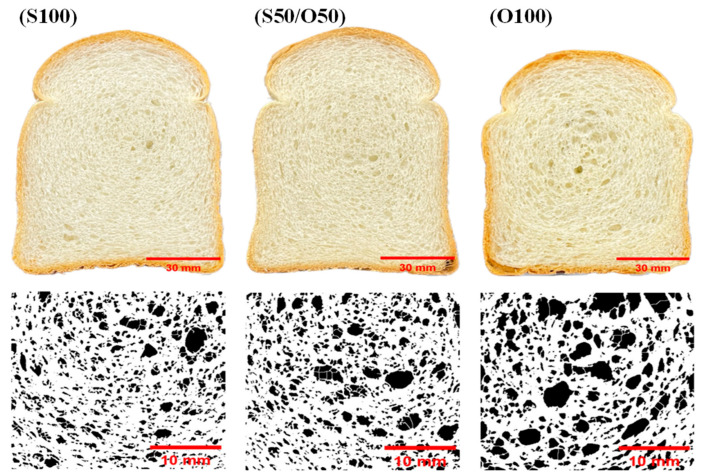
Cross sectional image analysis of crumb structure with different oleogel contents: (S100) control; (S50/O50) oleogel with 50% shortening substitution; (O100) oleogel with 100% shortening substitution.

**Table 1 foods-13-01678-t001:** Physicochemical quality characteristics of 0% shortening substitution (S100), 50% shortening substitution (S50/O50), and 100% shortening substitution (O100) of white pan bread prepared with shortening and oleogels.

	S100	S50/O50	O100
Baking loss (%)	15.83 ± 0.07 ^a^	15.74 ± 0.08 ^a^	13.42 ± 0.22 ^b^
Specific volume (ml/g)	5.32 ± 0.02 ^a^	5.35 ± 0.10 ^a^	4.85 ± 0.02 ^b^
Cell density (cells/cm^2^)	29.52 ± 0.24 ^a^	29.04 ± 0.17 ^a^	28.18 ± 0.17 ^b^
Average cell area (mm^2^)	0.54 ± 0.03 ^c^	0.66 ± 0.02 ^b^	1.08 ± 0.02 ^a^
Porosity (%)	16.03 ± 0.78 ^c^	19.16 ± 0.38 ^b^	30.53 ± 0.50 ^a^

All experiments were performed in triplicate. Data are shown as means ± standard deviation. Values with different letters in the same column differ significantly (*p* < 0.05).

**Table 2 foods-13-01678-t002:** Proximate analysis of 0% shortening substitution (S100), 50% shortening substitution (S50/O50), and 100% shortening substitution (O100) of white pan bread prepared with shortening and oleogels.

g/100 g	S100	S50/O50	O100
Crude protein	10.08 ± 0.03 ^a^	10.09 ± 0.06 ^a^	9.92 ± 0.05 ^b^
Crude fat	4.75 ± 0.35 ^a^	5.14 ± 0.11 ^a^	4.81 ± 0.01 ^a^
Ash	1.31 ± 0.16 ^a^	1.46 ± 0.01 ^a^	1.52 ± 0.01 ^a^
Moisture content (%)	33.50 ± 0.26 ^a^	33.63 ± 0.43 ^a^	35.26 ± 0.29 ^b^
Carbohydrate	52.10 ± 0.44 ^a^	51.70 ± 0.17 ^ab^	50.69 ± 0.53 ^b^
Calorie	291.70 ± 1.61 ^a^	293.30 ± 0.58 ^a^	286.70 ± 0.23 ^b^

All experiments were performed in triplicate. Data are shown as means ± standard deviation. Values with different letters in the same column differ significantly (*p* < 0.05).

**Table 3 foods-13-01678-t003:** Color analysis of 0% shortening substitution (S100), 50% shortening substitution (S50/O50), and 100% shortening substitution (O100) of white pan bread prepared with shortening and oleogels.

		S100	S50/O50	O100
Crust	L*	61.70 ± 0.21 ^b^	62.50 ± 0.01 ^a^	57.60 ± 0.30 ^c^
	a*	16.47 ± 0.02 ^b^	16.41 ± 0.05 ^b^	18.61 ± 0.08 ^a^
	b*	30.89 ± 0.07 ^b^	31.63 ± 0.09 ^a^	29.25 ± 0.01 ^c^
	BI (Browning index)	85.99	86.71	91.32
Crumb	L*	83.16 ± 0.06 ^b^	83.83 ± 0.11 ^a^	80.48 ± 0.34 ^c^
	a*	2.53 ± 0.03 ^a^	2.42 ± 0.02 ^b^	2.11 ± 0.01 ^c^
	b*	9.61 ± 0.08 ^c^	10.45 ± 0.04 ^b^	12.67 ± 0.13 ^a^

All experiments were performed in triplicate. Data are shown as means ± standard deviation. Values with different letters in the same column differ significantly (*p* < 0.05).

**Table 4 foods-13-01678-t004:** Texture profile analysis of 0% shortening substitution (S100), 50% shortening substitution (S50/O50), and 100% shortening substitution (O100) of white pan bread prepared with shortening and oleogels.

	S100	S50/O50	O100
Hardness, day 1 (N)	5.18 ± 0.03 ^b^	4.40 ± 0.02 ^c^	8.43 ± 0.04 ^a^
Hardness, day 2 (N)	7.82 ± 0.04 ^b^	5.86 ± 0.04 ^c^	12.23 ± 0.01 ^a^
Hardness, day 4 (N)	10.34 ± 0.02 ^b^	7.83 ± 0.07 ^c^	14.00 ± 0.11 ^a^
Springiness, day 1 (mm)	9.64 ± 0.01 ^a^	9.71 ± 0.02 ^a^	9.64 ± 0.20 ^a^
Cohesiveness, day 1	0.67 ± 0.01 ^a^	0.68 ± 0.01 ^a^	0.65 ± 0.01 ^a^
Chewiness, day 1 (J)	34.54 ± 0.45 ^b^	29.17 ± 0.65 ^c^	53.08 ± 0.49 ^a^

All experiments were performed in triplicate. Data are shown as means ± standard deviation. Values with different letters in the same column differ significantly (*p* < 0.05).

**Table 5 foods-13-01678-t005:** Fatty acid compositions of 0% shortening substitution (S100), 50% shortening substitution (S50/O50), and 100% shortening substitution (O100) of white pan bread prepared with shortening and oleogels.

Fatty Acid (g/100 g)	S100	S50/O50	O100
C4:0	0.38 ± 0.00 ^a^	0.39 ± 0.01 ^a^	0.42 ± 0.02 ^a^
C6:0	0.26 ± 0.01 ^a^	0.25 ± 0.01 ^a^	0.26 ± 0.01 ^a^
C8:0	0.20 ± 0.01 ^a^	0.18 ± 0.01 ^b^	0.17 ± 0.01 ^c^
C10:0	0.41 ± 0.00 ^a^	0.38 ± 0.01 ^b^	0.37 ± 0.01 ^c^
C12:0	0.92 ± 0.01 ^a^	0.71 ± 0.01 ^b^	0.52 ± 0.01 ^c^
C14:0	2.51 ± 0.00 ^a^	1.90 ± 0.01 ^b^	1.31 ± 0.01 ^c^
C16:0	37.66 ± 0.02 ^a^	26.68 ± 0.20 ^b^	15.78 ± 0.15 ^c^
C16:1	0.82 ± 0.01 ^c^	0.88 ± 0.01 ^b^	0.96 ± 0.01 ^a^
C18:0	7.43 ± 0.00 ^a^	5.96 ± 0.04 ^b^	4.61 ± 0.10 ^c^
Trans C18:1	1.35 ± 0.01 ^a^	0.83 ± 0.03 ^b^	0.33 ± 0.03 ^c^
C18:1	33.44 ± 0.02 ^c^	47.48 ± 0.29 ^b^	61.24 ± 0.15 ^a^
Trans C18:2	0.38 ± 0.01 ^a^	0.24 ± 0.00 ^b^	0.09 ± 0.00 ^c^
C18:2	13.25 ± 0.04 ^a^	12.86 ± 0.01 ^b^	12.42 ± 0.09 ^c^
C18:3	0.49 ± 0.01 ^c^	0.68 ± 0.01 ^b^	0.88 ± 0.01 ^a^
C20:0	0.29 ± 0.01 ^c^	0.33 ± 0.01 ^b^	0.37 ± 0.00 ^a^
C20:1	0.21 ± 0.00 ^c^	0.23 ± 0.01 ^b^	0.26 ± 0.01 ^a^
SFA (%)	50.06 ± 0.03 ^a^	36.79 ± 0.29 ^b^	23.80 ± 0.21 ^c^
USFA (%)	48.20 ± 0.05 ^c^	62.13 ± 0.30 ^b^	75.76 ± 0.25 ^a^
TFA (%)	1.73 ± 0.01 ^a^	1.07 ± 0.03 ^b^	0.42 ± 0.03 ^c^
SFA/USFA	1.04 ± 0.00 ^a^	0.59 ± 0.01 ^b^	0.31 ± 0.01 ^c^

All experiments were performed in triplicate. Data are shown as means ± standard deviation. Values with different letters in the same column differ significantly (*p* < 0.05).

## Data Availability

The original contributions presented in the study are included in the article, further inquiries can be directed to the corresponding author.

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
