# Peer review of "Development of Oleogel-Based Fat Replacer and Its Application in Pan Bread Making"

_foods, 2024, doi:10.3390/foods13111678_

Round 1

Reviewer 1 Report

Comments and Suggestions for Authors

In this manuscript oleogels were prepared from hydroxypropyl methylcellulose (HPMC), xanthan gum (XG), and olive oil and then they were use instead of shortening in the preparation of bread. It is an interesting subject. The work has been done in a smooth way, but there is a need for a revision.

Abstract should be with some data and informative. Line 18, potential reductions; line 19, increasing substitution ratio-noticeable decline, line 20, increase in unsaturated these statements should be with data. Line 23, quality characteristics, should be defined which quality parameters?

Line 72 and also in whole manuscript, please give the references with following the journal format, [18-20] [21,22].

Shortening could be produced by hydrogenation and also intersterfication. Please add it in the text.

Please indicate the type of olive oil, refined, virgin?

Figure 1, please describe the abbreviation in the figures, DW?

Line 200, ... does not have a major impact on bread quality., but in the next discussion part for example lines 268- 294, there is a discussion about the oleogel significant effects on quality.

Figure 3 is not complete and should be replaced.

Comments on the Quality of English Language

English needs a revision to be smooth and easy to follow for readers, for example, line 284-285, These findings highlight that S50/O50 showed a reduced hardness increase rate of 35.46% (1.83) compared to S100< this is an unclear statement. There is several writing like this sentence in the manuscript, please recheck and revise them.

Line 286 and 287, there is no results about retrogradation and chewing, so it can be declared in the text, may be it is fine if you give it as discussion with references from already published papers.

Author Response

Thanks for your comprehensive review for our manuscript. We appreciate your feedback about our manuscript and carefully answered your comments or questions. We would like to provide some explanations as follows.

  1. Abstract should be with some data and informative. Line 18, potential reductions; line 19, increasing substitution ratio-noticeable decline, line 20, increase in unsaturated these statements should be with data. Line 23, quality characteristics, should be defined which quality parameters?
  • As you recommended, we added specific data in abstract.
  1. Line 72 and also in whole manuscript, please give the references with following the journal format, [18-20] [21,22].
  • We corrected
  1. Shortening could be produced by hydrogenation and also intersterfication. Please add it in the text.
  • We added it in the text.
  1. Please indicate the type of olive oil, refined, virgin?
  • We corrected in manuscript
  1. Figure 1, please describe the abbreviation in the figures, DW?
  • We deleted, reviewer #2 recommended to delete figure 1 and 2.
  1. Line 200, ... does not have a major impact on bread quality., but in the next discussion part for example lines 268- 294, there is a discussion about the oleogel significant effects on quality.
  • We corrected
  1. Figure 3 is not complete and should be replaced.
  • We corrected Figure 3
  1. Comments on the Quality of English Language
    English needs a revision to be smooth and easy to follow for readers, for example, line 284-285, These findings highlight that S50/O50 showed a reduced hardness increase rate of 35.46% (1.83) compared to S100< this is an unclear statement. There is several writing like this sentence in the manuscript, please recheck and revise them.
  • We corrected sentences
  1. Line 286 and 287, there is no results about retrogradation and chewing, so it can be declared in the text, may be it is fine if you give it as discussion with references from already published papers.
  • We have reviewed and incorporated the TPA results into the text, providing interpretation and discussion of the springiness, chewiness, and cohesiveness measurements.

Reviewer 2 Report

Comments and Suggestions for Authors

The authors had an efficient task in their research and obtained beneficial results regarding the addition of oleogel in bread production.

I believe that the manuscript is written correctly and still requires a reorganization of the structure and some corrections. In my opinion, the research was carried out correctly and the main goal of the research was achieved - the addition of olegel composed of olive oil, HPMC, and XG can be used in bread production as an alternative to traditional solid fats.
The following issues/points should be taken into consideration (in PDF file).

Author Response

Introduction

  1. The introduction was written correctly and was based on currently available literature. However, it requires emphasising what new contributions your research brings to the world of science. Please add one or two sentences about the novelty of your research.
  • We added sentences
  1. Line 68: place the citation in one square bracket, giving the digits after the decimal point [12,17], Materials and methods Please change the subsection numbers.
    • We corrected
  2. Subchapter 2.3. is too long. Subsections, for example, 2.3.4, 2.3.5, or 2.3.6, should constitute another subsection, e.g., 2.5, 2.6 or 2.7. Please think about and reorganize it.
    • We corrected
  3. LINE 93: please fill in the type of flour, it was wheat flour?
    • We corrected. - wheat flour.
  4. There is no need to include the block diagrams presented in Figures 1 and 2 as it has been clearly described in the text. I recommend removing Figures 1 and 2.
    • As recommended, we removed figures 1 and 2.
  5. LINE 140: I would recommend using appropriate symbols in the formula, placing an appropriate explanation below and providing the unit in square brackets.
    • We corrected
  6. LINE 144: Please list what texture parameters were determined and their units.
    • We added texture parameters and units.

Results and Discussion

  1. Subchapters to a chapter Results and Discussion should be titled by the order given in the methodology.
    • We corrected
  2. Please take this into account because the titles are given out of order. Please see line 190, line 310.
    • We corrected
  3. LINE 196: Figure ??? something is missing, not a complete thought
    • We corrected
  4. LINE 203: Please quote appropriately, giving the author's name and the publication number in square brackets (the same LINE 274).
    • We corrected
  5. In my opinion, the discussion on color parameters and fatty acids needs to be improved (line 246, 407).
    • We corrected color and fatty acids section discussion.
  6. Namely, the results in this subsection should be compared with literature data in which the addition of other oleogel was used.
    • We added references.
  7. Please correct that the color parameters L*, a*, b* have the * sign in the text, this will improve the readability of the results obtained
    • We corrected
  8. The conclusions are consistent and supported by evidence. The References were selected correctly. In addition, I have some minor comments regarding the Figures, LINE 216: Figure 3. Please insert a photo with appropriate zoom so that 3 whole loaves can be seen.
    • We corrected
  9. LINE 303: Please consider removing Figure 6 because the data was previously presented in Table 5.
    • We removed figure 6.

Reviewer 3 Report

Comments and Suggestions for Authors

The presented manuscript is very valuable, however, it contains certain inconsistencies that need to be clarified or corrected.

Keyword "fat replacement" repetition from the title.

The Introduction provides a well-composed overview of the most important information related to the manuscript's topic.

The Reference list of scientific publications seems to be sufficient. The only reservations may concern the timeliness of the data. The manuscript mainly relies on sources up to the year 2020, with only one reference from 2021. I suggest incorporating 2-4 of the latest references from the years 2020-2024.

Lines 90-95 describe the quantity of used raw materials, which is then reiterated in Table 1 (repetition of information).

In section 2.3. Preparation of white pan bread with oleogel – there is no information about the quantity of baked loaves. Was the baking process repeated?; Line 119, the baking temperature differs from that in Figure 2.

In section 3.3.3. "to determine metrics such as cell density (cells/cm²), average cell area (ratio of 150 pore area to the number of cells), and porosity (ratio of pore area to 3 cm²)." – the authors only provide results related to porosity. The provided porosity values, in comparison with the applied determination methodology, raise doubts. If the crumb porosity was only 0.15%, it means that it was a very dense crumb, almost without porosity – which is not confirmed by the images in Figure 4. Additionally, please include data for the other indicators described in the methodology.

 Color analysis of oleogel bread at line 166 mentions Equation 1, but Equation 1 describes baking loss.

I suggest changing the order of tables 2 and 3. The authors describe the physical properties of the studied bread as the first ones, which are presented in Table 3. Additionally, there is no reference in the text to the chemical composition.

Graph 6 is a repetition of data from Table 5.

Please supplement information regarding TPA parameters other than hardness – there is no information in which day the data presented in the table were collected. Why were only bread hardness determined in subsequent days?

Inferring the extent of retrogradation based solely on hardness is an overinterpretation. Changes in bread hardness over storage time are undoubtedly caused by retrogradation, but not exclusively. Other factors are also significant here, such as changes in crumb moisture content. Please correct this overinterpretation both in the Results and Discussion section as well as in the Conclusion.

The authors present TPA results in the table but do not refer to them in the text or interpret them (Springiness, chewiness, cohesiveness).

More comments have been included in the attached file.

Author Response

  1. The Reference list of scientific publications seems to be sufficient. The only reservations may concern the timeliness of the data. The manuscript mainly relies on sources up to the year 2020, with only one reference from 2021. I suggest incorporating 2-4 of the latest references from the years 2020-2024.
  • We added references.
  1. Lines 90-95 describe the quantity of used raw materials, which is then reiterated in Table 1 (repetition of information).
  • We removed table 1.
  1. In section 2.3. Preparation of white pan bread with oleogel – there is no information about the quantity of baked loaves.
  • We included information about the quantity of baked loaves in section 2.3.
  1. Was the baking process repeated? Line 119, the baking temperature differs from that in Figure 2.
  • We've noted the discrepancy and made the necessary adjustments for consistency.
  1. In section 3.3.3. "to determine metrics such as cell density (cells/cm²), average cell area (ratio of 150 pore area to the number of cells), and porosity (ratio of pore area to 3 cm²)." – the authors only provide results related to porosity. The provided porosity values, in comparison with the applied determination methodology, raise doubts. If the crumb porosity was only 0.15%, it means that it was a very dense crumb, almost without porosity – which is not confirmed by the images in Figure 4. Additionally, please include data for the other indicators described in the methodology.
  • We apologize for the oversight in our response. We have included the requested data for cell density and average cell area, in addition to revising the porosity values for consistency with the applied methodology.
  1. Color analysis of oleogel bread at line 166 mentions Equation 1, but Equation 1 describes baking loss.
  • We have corrected the reference to Equation 1 in the color analysis of oleogel bread at line 166.
  1. I suggest changing the order of tables 2 and 3.
  • We changed the order and numbering of the tables as suggested.
  1. The authors describe the physical properties of the studied bread as the first ones, which are presented in Table 3. Additionally, there is no reference in the text to the chemical composition.
  • We did not mention the chemical composition in the text because, except for the moisture content, there were no significant differences in the general composition.
  1. Graph 6 is a repetition of data from Table 5.
  • We removed figure 6.
  1. Please supplement information regarding TPA parameters other than hardness
  • We added information regarding TPA parameters other than hardness.
  1. there is no information in which day the data presented in the table were collected.
  • We corrected the data to include information on the specific days when they were collected.
  1. Why were only bread hardness determined in subsequent days?
  • We added information on chewiness and moisture content, along with relevant references.
  1. Inferring the extent of retrogradation based solely on hardness is an overinterpretation. Changes in bread hardness over storage time are undoubtedly caused by retrogradation, but not exclusively. Other factors are also significant here, such as changes in crumb moisture content. Please correct this overinterpretation both in the Results and Discussion section as well as in the Conclusion.
  • We corrected this by adding information on chewiness and moisture content, along with relevant references, to provide a more comprehensive interpretation of the changes in bread hardness over storage time.
  1. The authors present TPA results in the table but do not refer to them in the text or interpret them (Springiness, chewiness, cohesiveness).
  • We have reviewed and incorporated the TPA results into the text, providing interpretation and discussion of the springiness, chewiness, and cohesiveness measurements.
  1. More comments have been included in the attached file.
  • We have reviewed the attached file and made the necessary corrections
